# A dynamic model to sustain the spark: How do network coordinators in Dutch healthcare networks maintain network commitment?

Manon Roest[1]*, Doris van der Smissen[1], Remco Mannak[2], Leon Oerlemans[2,3], Anne Marie Weggelaar-Jansen[1]

1 Tranzo, Tilburg School of Social and Behavioral Sciences, Tilburg University, Tilburg, Netherlands, 2 Organization Studies, Tilburg School of Social and Behavioral Sciences, Tilburg University, Tilburg, Netherlands, 3 Engineering and Technology Management, University of Pretoria, Pretoria, South Africa

* m.j.roest@tilburguniversity.edu

## Abstract

### Context

In the Netherlands inter-organizational networks have been established to drive long-term healthcare improvement on a regional and national scale. Managing healthcare networks is challenging, especially in sustaining participants' active engagement. This study aims to empirically explore why participants' network commitment declined and how network commitment was managed by network coordinators.

### Methods

We conducted an exploratory qualitative study using purposive sampling we held semi-structured interviews with representatives from 18 Dutch healthcare improvement networks.

### Results

Respondents mentioned that allocated time and continued participant commitment was required to reach their goals. A key challenge was the decline in participant commitment over time due to misalignment of goals and needs, slow progress, lack of reciprocity, and resistance to change at the organizational level. Approaches to manage this were collaborative reassessment of goals and needs, expectation management, setting preconditions and fostering personal relationships.

### Discussion

Our findings highlight the importance of strategic timing of managerial approaches and their interdependencies with social mechanisms. These insights contribute theoretically to Ring and Van de Ven's framework on developmental processes of

**Data availability statement:** All relevant data are within the paper and its Supporting information files. We have attached a code tree in English and the codebook in Dutch. The qualitative data in this study are available in the DataverseNL repository, under DOI: https://doi.org/10.34894/SHVP4Y. The dataset is publicly accessible and can be accessed by interested researchers.

**Funding:** co-Authors AW and DvS received funding from the the Dutch Ministery of Health via ZonMW Grant number: 90000743 URL: https://www.zonmw.nl/nl/nieuws/patientveiligheid-vergroten-met-safety-ii-11-onderzoeken-van-start The subsidizing party did not play any role in the study design, data collection, data analysis, decision to publish or preparation of the manuscript.

**Competing interests:** The authors have declared that no competing interests exist.

cooperative interorganizational relationships and offer practical guidance for network managers aiming to maintain long-term commitment.

## Introduction

Healthcare systems worldwide are under increasing pressure due to a variety of challenges, including a rising demand for healthcare services and their quality, cost containment, improving patient experiences, and ensuring a sustainable work environment for healthcare professionals [1,2]. Due to their high complexity, these challenges cannot be addressed by individual organizations. Instead, they require collaborative approaches that bring together multiple organizations and stakeholders to develop integrated, sustainable solutions [3,4]. Consequently, inter-organizational network collaboration has become increasingly prevalent in public and private sectors [5]. They serve as platforms for joint problem-solving and provide channels for knowledge exchange and policy implementation [6–8].

The Netherlands is no exception to these challenges and this collaborative trend. Various policy initiatives and structural reforms have encouraged the development of healthcare networks that facilitate integrated care service delivery to enhance patient safety and organize care closer to home [9,10]. A practical example is a regional network focused on optimizing the care pathway for elderly patients living in vulnerable circumstances at home. Such an initiative demands collaboration between general practitioners, home care providers, emergency services, welfare, and local policymakers. The Dutch Ministry of Health offers funding to nation-wide programs to accelerate improvements via multi-organizational partnerships in networks aiming for shared learning [11]. For example, the Healthy and Active Living Agreement (GALA) or the Integral Care Agreement [12]. Achieving such improvement goals requires the long-term active involvement of a wide range of stakeholders working in close collaboration. These actors must jointly assess existing practices, identify opportunities for improvement, and implement changes across multiple organizations. This change trajectory is inherently complex, as stakeholders' priorities and perspectives often differ and evolve over time. Therefore, this complexity requires more than an initial agreement; it depends on sustained commitment and shared problem ownership to drive and maintain meaningful change. In this paper we explore networks aiming to improve healthcare.

### Inter-organizational networks and network commitment

Inter-organizational networks, including healthcare networks, enable organizations to pool diverse resources, share knowledge, and work toward a common goal [13]. Research has shown that networks can improve care quality by enhancing care coordination through collaborative professional relationships [14]. Also, networks enhance physician performance by fostering social capital [15]. On a national level, networks can play a crucial role in implementing national policies, facilitating major service reconfiguration, and securing clinical engagement [16]. For networks to be effective and attain their goals – particularly in the context of healthcare improvement-,

networks must remain stable and foster sustainable collaboration [17,18]. Sustainable collaboration relies on a foundation of shared values, trust, and effective communication, particularly supported by senior leadership [19]. Additionally, Provan and Kenis emphasize the importance of goal congruence, trust, and governance in maintaining networks [8]. An additional critical factor for network sustainability is network commitment, which can be understood as an organization's intention to stay in the network and actively contribute to make the collaboration a success [20]. Network commitment encompasses an active engagement and psychological attachment [21]. High network commitment levels foster the quality of decision-making and maximize collaborative efforts and success [18–20]. Moreover, it reinforces trust among participants, strengthens interpersonal and inter-organizational collaboration [21].

The importance of network commitment for inter-organizational collaborative structures relates to one of the fundamental characteristics of inter-organizational networks: dispersed hierarchy and control [22]. In inter-organizational networks, decision-making is shared between legally autonomous entities and not unified within one organizational structure. Following Clarke [18], network commitment is, therefore, regarded as a coordinating mechanism to compensate for the lack of behavioral (process) control and traditional intra-organizational hierarchy.

Maintaining commitment at adequate levels among participants is far from easy. Causes for commitment decline include insufficient time and resources to execute tasks as needed, an imbalance between the network and organizational goals [32], and a too centralized decision-making. This is reinforced when participation is voluntary, loosely coupled, not contractually enforced [23], and simultaneously network goals are long-term [24]. In this paper we elaborate on network commitment of healthcare networks with such long-term goals.

## Managing networks

Managing healthcare networks has proven to be challenging, as it requires continuous coordination among multiple stakeholders, including healthcare professionals, hospital managers, insurers, and policymakers [10]. Network managers need to balance competing interests, allocate resources efficiently, and align long-term strategic goals [25]. Several risks can threaten network stability and may lead to network failure, such as micro-management, lack of institutional flexibility, limited collective strategy, insufficient knowledge sharing, and prioritization of individual interests over collective goals [26,27]. Given these challenges and risks, it is crucial to get a clear insight and understanding of the contributing factors to maintaining commitment and achieving the network goals [20]. While existing research predominantly has focused on network effectiveness, including governance structures, role clarity, and shared purpose [25] there remains a limited understanding of how to sustain network commitment over time [28]. This is particularly relevant because a deeper understanding of how networks function and evolve is vital for network managers aiming to maintain engagement [14].

This study aims to empirically explore if, and why the network commitment of participants declined and how network commitment was managed by network coordinators in Dutch healthcare improvement networks. By exploring the factors influencing the development of network commitment and the approaches coordinators used to maintain engagement, this research will provide insight into the management of network commitment within the Dutch healthcare context. Ultimately, the findings will contribute to actionable recommendations for network managers seeking to enhance participant engagement and ensure network sustainability.

## Methods

To gain an in-depth understanding of how network commitment may decline, and how this is managed by network coordinators in Dutch healthcare improvement networks, we conducted an exploratory, qualitative study using semi-structured interviews. This study was part of a commissioned research project called "Time to Connect", which aimed to enhance network collaboration to enhance patient safety in Dutch hospitals and private clinics. We followed the consolidated Criteria for Reporting Qualitative Research (COREQ) to report the study (see S1 Appendix). Ethical approval was secured from the School's Ethics Review Board before study commenced [blinded for peer review]. Data storage is compliant with the GDPR.

**"Time to connect"**

The nationwide patient safety program "Time to Connect" focused on some of the most prevalent safety risks in the Netherlands, including anticoagulation care, frail elderly care, and multidisciplinary care for complex cases. The program fostered resilient behavior of healthcare professionals by offering collaborative learning and transference of positive deviance practices through networks in which reflection-and-improvement sessions were facilitated. One example of inter-organizational collaboration aimed for quality improvement was the establishment of a network to optimize the care (pathway) for frail elderly. This network required the involvement of general practitioners, care and welfare providers, emergency services, and local policymakers on organizational and municipality level. The reported study was undertaken to support the development of the 'Time to Connect' program. By studying the evolvement of network commitment in other national programs, insights were gained into how to establish vibrant and sustainable networks.

**Inclusion criteria**

To ensure a diverse and representative sample, we used a maximum variation sampling approach which is one of the purposive sampling methods described by Patton (2002) [29]. We started by mapping existing Dutch healthcare networks that operate(d) at either regional or national level and aim(ed) for improving healthcare. The network improvement goals varied and included enhancing patient safety for a specific patient group, strengthening shared-decision making for patients, or improving operational management (e.g., reduction of waiting times, development of care pathways etc.). To ensure a comprehensive overview, we sampled networks that varied in size, participation requirements, funding, leadership and governance modes, and anticipated outcomes. Exclusion criteria included networks comprising fewer than three organizations, networks focusing solely on knowledge exchange without broader improvement goals, short-term inter-organizational projects, or newly established networks.

**Data collection**

The recruitment period spanned from February to September 2023. Interviewees were required to hold a formal leadership or managerial role in the networks (e.g., director, program leader, project manager, or coordinator), ensuring they had direct managerial experience with network functioning. Additionally, we interviewed network advisors who had previously participated in networks that suited our inclusion criteria. S1 Table provides an overview of the respondents' characteristics and network-specific attributes, including types of professionals involved as participants, funding structures, and governance models. Contact persons of eligible networks were contacted by email inviting them to participate. The invitation outlined the purpose of the study and included an informed consent form. Recruitment continued until thematic saturation was reached, resulting in a total of 18 interviews with responsible managers of Dutch healthcare networks. One network program manager declined participation, due to maternity leave.

Two researchers (DvdS, MR) conducted the semi-structured interviews online, using Microsoft Teams. The interviews followed a predefined topic guide (see S2 Appendix) that stemmed from a literature search on barriers and facilitators for healthcare improvement in inter-organizational networks. Before each interview, respondents provided verbal informed consent, which was audio-recorded. Interviews were automatically transcribed verbatim using the transcription tool in Microsoft Teams. Afterward, transcripts were manually checked for accuracy and pseudonymized by removing personal identifiers.

**Data analyses**

Transcripts were analyzed using inductive thematic content analysis following Braun and Clarke's six-step framework [30]. The analysis was conducted using the analysis software Atlas.ti.24. First, one researcher (DvdS) read, and openly coded the transcripts of four interviews. These initial codes were used to form an initial coding tree consisting of four preliminary

                                                          

themes and (sub)concepts By axial coding, the remaining interviews were coded using the coding tree by assigning axial codes and by adding new open codes, concepts and sub-concepts. Throughout the entire analysis process, researchers from different disciplines (DvdS, MR, AWJ) discussed the coding tree up to consensus. Gradually, in close collaboration we finalized the coding tree (See S3 Appendix for the code tree and S4 Appendix for the codebook). We validated the coding tree during a session with employees of the "Time to Connect" program. After 18 interviews, thematic saturation was reached.

## Results

The representatives of the networks aiming for healthcare improvement all mentioned that their network required financial and personnel resources, but most importantly time. As one respondent noted:

> "*Arranging a network is so complex. It's not just the issues that require collaboration, but the collaboration itself takes time, as well as organizing it.*" (*P7: Network coordinator*)

Coordination of network management tasks was often entrusted to an appointed project leader, also referred to as a network coordinator. Depending on the nature of the network, this coordinator could be a medical professional, a manager of one of the participating organizations, or an external party. This person facilitated connections between organizations, motivated participant contribution, and disseminated valuable lessons learned among participants.

A key network management challenge mentioned by these coordinators was the declining commitment of participants over time. Even though commitment to the network was present at the start of the network, over time, network participants often became less engaged. In the following section, we elaborate on 1) the main reasons for participants' decline in network commitment; and 2) the strategies or interventions used by the network coordinator to counteract this decline or to restore commitment.

### Reasons for declining network commitment of participants

Generally, the initial commitment of participants to actively engage within the network was strong, often driven by an intrinsic sense of urgency for healthcare improvement and a genuine desire to improve the care quality and safety. Additionally, some network participants were motivated by funding opportunities, while others engaged due to external requirements, such as mandates from healthcare insurers or government policies. However, in many cases, network commitment eventually declined.

One of the main reasons for this decline was the misalignment between the network's objectives and the evolving needs of its participants. Although a common goal was agreed upon at the start of the network to foster alignment and engagement, discrepancies emerged as the network progressed. As the organizational priorities and expectations shifted, some participants experienced a growing disconnect between their needs and the overarching network aims, ultimately leading to decreased involvement.

A second reason for network commitment decline arose when participants did not experience immediate progress toward reaching the defined goals, particularly in networks with inherently lengthy processes. The slow pace of progress, often due to complex, multidisciplinary challenges, led to declining motivation and commitment over time. This was particularly evident among network members who got involved in the early stages of the network. In contrast, participants who joined an already operational network often experienced quicker results, reinforcing their commitment to the network. The following quote illustrates this:

> "*So, those who joined later saw results immediately, which is great. Naturally, they were highly motivated by this, whereas people from the first hour, the initial five healthcare organizations had to work quite hard to achieve these*

*results. At some point, you realize you need to invest extra energy to keep them [early participants] motivated. It's a long-term process, and it needs speed, but ultimately you get a better outcome together than going at it alone. You make more progress with collaboration, really. Still, the first healthcare organization had to go on a long journey. Now we have the results it's okay, but in that preceding phase, it was a bit challenging."* (*P5: Managing director of a network*)

A third key reason for declining commitment among network participants was a lack of reciprocity in the collaboration within the network. Many participants joined the network because of the opportunity to exchange knowledge, such as sharing best practices, experiences, and insights. This knowledge exchange was considered crucial for initial commitment. However, over time the knowledge sharing between participants proved to be unbalanced. Consequently, the overall commitment weakened. When participants experienced that sharing their knowledge was not answered by getting something in return, they became less inclined to actively continue their sharing behavior. The quote below illustrates how ongoing reciprocity is needed for commitment:

*"When you can find those things elsewhere, [active participation] becomes redundant, so to speak. Well then, perhaps it's less worthwhile to invest effort in it, because, again, a network is built on reciprocity. If you invest time in bringing about something and taking something out of it, you need ongoing reciprocity. Otherwise, at some point, the costs outweigh the benefits."* (*P8: Network coordinator*)

This lack of reciprocal engagement was also present when some participants joined the network solely to acquire insights for their organization or to secure funding, without actively contributing to the network. This behavior was sometimes attributed to opportunism, or to work-related pressure, as network participation is often an additional responsibility and task on top of the existing workload.

Lastly, resistance to change within participating organizations contributed to declining network commitment. While organizations were initially eager to collaborate on the objective of improving care, challenges arose when new working practices developed, which required changes within the organizations. In some cases, employees within participating organizations resisted change due to increased workloads, disruption of routines, or financial constraints. This internal pushback sometimes led participants to become less engaged.

## Managerial approaches for dealing with network commitment decline

Network coordinators used various approaches to manage their network and to prevent a decline in participants' network commitment. One of the approaches was regular assessment of the network objectives to ensure they remained aligned with participants' evolving needs. To identify rising discrepancies between the network's goals and the needs of its members, network managers conducted periodic evaluations, either through surveys or in-person discussions. When misalignments were detected, adjustments could be made to realign the objectives and maintain commitment. This quote exemplifies that adjusting objectives is needed to solve a problem, but also to keep participants committed:

*"Our goal is not to just keep the network active. We're really trying to solve a problem. But when it comes to keeping a network effective over time? Yeah, how do we do that? I believe it's by not adhering too rigidly to the chosen format or sticking to it indefinitely, right? [...] With everything you do and every time you come together, just ask critically, why are we doing this? So, you don't end up with people sitting at meetings only because they've been doing it for ten years already and everybody has lost interest, but yeah, it's on the agenda."* (*P10: Managing director of a network*)

Another approach to maintain commitment was managing participants' expectations, particularly for early-stage participants. By clarifying and stressing at the outset, that goal attainment takes time, and yield long-term benefits, participants

are less likely to become discouraged when immediate results are not emerging. It was helpful to explain that participating can ultimately be beneficial to both the network and the participating organizations. In addition to expectation management, highlighting successes – even incremental ones – was an effective way to reinforce or re-establish motivation. This reassured participants that their contributions were making a difference, strengthening their willingness to stay engaged.

Ensuring reciprocity within the network was another crucial approach. Network coordinators employed various measures to foster mutual knowledge sharing and shared responsibility. Some networks set certain preconditions for participation, such as expected contributions (e.g., financial, in-kind, free flow of knowledge) or mandatory attendance at key meetings. Others enhanced engagement by rotating network meeting hosts, encouraging participants to suggest agenda topics and thereby fostering a sense of ownership by every member. The reciprocal engagement was advanced by trust-building among the participants. To build this trust, it was crucial to have transparency about individual benefits of participation, as well as the potential collaboration challenges.

A personal approach of the network coordinator towards the network participants and fostering inter-organizational connections were also needed when resistance to change arose at the organizational level. Creating and maintaining personal connections helped to create a form of accountability and ownership for continued collaboration. Personal connections were mainly established during social events, hosted by the network managers.

*"I reckon the most important aspect is that we know who's who and what we all do. So, we're constantly getting a clearer idea of people, and we keep on getting better at finding each other. That's really how a network works. You can connect as organizations, but if you don't know how to find the right person or don't have their number, or you don't see each other often enough, then too little actually happens. It comes down to, once you know someone well, you can rely on them. You must pay extra attention [to personal contacts], organize extra meetings, almost informally. We need this to actually achieve things."* (*P6a: Network researcher*)

Furthermore, involving senior leadership, such as the participating organizations' board members, proved to be a crucial strategy for overcoming organizational resistance to change as well. As board members often played a key role in decision-making processes, this higher-level support helped to prevent resistance and disengagement.

*"If you look at the organizational level, what is helpful, is that we have a well-structured board that is truly committed to what we are doing in our pilot and actively promote it within their own hospital. That's one important factor. However, of course, it is not enough to only have engagement at that level. You see that it takes time to also embed this awareness at the lower levels—to really get it into people's minds that there is something like [network name] and that they can benefit from it"* (*P5: Managing director of a network*)

## Discussion

Collaboration in networks of healthcare providers is primarily driven by the desire to improve healthcare delivery or care quality and safety [31]. With these objectives, maintaining commitment over time is essential for sustaining collective efforts, attaining common goals, and ensuring continued progress [32–34]. While previous literature has acknowledged the importance of network commitment [21,35,36] and identified several influential factors such as trust, shared purpose, and governance structures [21,25,33,37–39], there remains limited empirical insight into how network coordinators manage network commitment in practice. This study addressed this gap by exploring how network coordinators in Dutch healthcare improvement networks approached the challenge of declining network commitment. Table 1 provides an overview of the identified reasons for declining network commitment, alongside the corresponding approaches used to address them.

**Table 1. Reasons for declining network commitment and managerial approaches.**

| Reason for declining network commitment | |
|---|---|
| R1 | Rising discrepancies between network objectives and participants' needs |
| R2 | The slow pace of progress |
| R3 | The lack of reciprocal engagement |
| R4 | Resistance to change at the organizational level |
| **Managerial approach** | |
| A1 | Identify discrepancies through periodic evaluations and adjust objectives if necessary |
| A2 | Manage the expectations of participants and celebrate (small) successes |
| A3 | Set preconditions to foster equal contribution, create ownership and build trust among participants |
| A4 | Include senior leadership of the participating organization and foster personal connections and |

To position and demarcate this study's theoretical contributions, we make use of relevant review studies that recently have been published in the field of healthcare networks. The scoping review by Carmone et al [40], 2020 offers a thorough conceptualization of healthcare networks and a mapping of their activity domains. The healthcare networks investigated address complex problems, all aiming for system wide changes, which fall in the so-called 'learning and adaptation' domain. The relevance of this domain is acknowledged; still, this area is understudied, as this scoping review found the lowest number of insights in this domain. This observation makes this study of healthcare networks dealing with complex problems therefore timely and relevant. Additionally, a systematic literature review by Van der Weert [41], showed the fragmented nature of the scientific insights on healthcare networks and the mixed empirical findings on the relationship between network structures and governance modes on the one hand, and healthcare network performance at different levels on the other hand. The authors [41] observed that the literature predominantly focuses on healthcare networks' structural aspects, whereas intermediate processes such as involvement and commitment attract far less scholarly attention. Furthermore, network commitment already captivated academic curiosity in the past. Following the collaborative governance framework [42], network commitment is regarded as an integral part of collaborative dynamics. More specifically, network commitment is conceptualized as a dimension of shared motivation together with mutual trust and understanding, and internal legitimacy. These collaborative dynamics are the engine that drives Collaborative Governance Regimes, such as healthcare networks, to collaborative action. However, Belrhiti et al. [42], found that most studies concentrate on commitment building at the early formation stage of healthcare networks. How network coordinators manage network commitment over time is, however, an unexplored area, which need scholarly attention in light of problem complexity and the stable and sustainable engagement required to address it. By providing an overview over the reasons for declining network commitment and the approaches used to address these, we offer new insights.

Our theoretical contribution builds on and extends Ring and Van de Ven's theoretical framework concerning the developmental processes of cooperative interorganizational relationships [43]. The framework outlines three cyclical stages of interorganizational cooperation. Stage one, negotiations, is when the expectations, roles, and resources are aligned. Stage two, commitments, includes choosing the approach, governance structure, and trust-building. The third stage, execution, involves executing these commitments through actions. Given the central and enduring role that this renowned framework attributes to network commitment, makes the framework particularly suitable for conveying the study's theoretical contribution.

Fig 1 emphasizes the interdependency between managerial actions in different stages and their social mechanisms. We argue that network participants' needs may evolve, potentially leading to misalignment (R1) with the network goals [44]. This affects goal attainment, network effectiveness, and weakens network commitment, as even the perception of misalignments creates dissatisfaction among participants with the network and reduces commitment [8,36,37,45]. In the

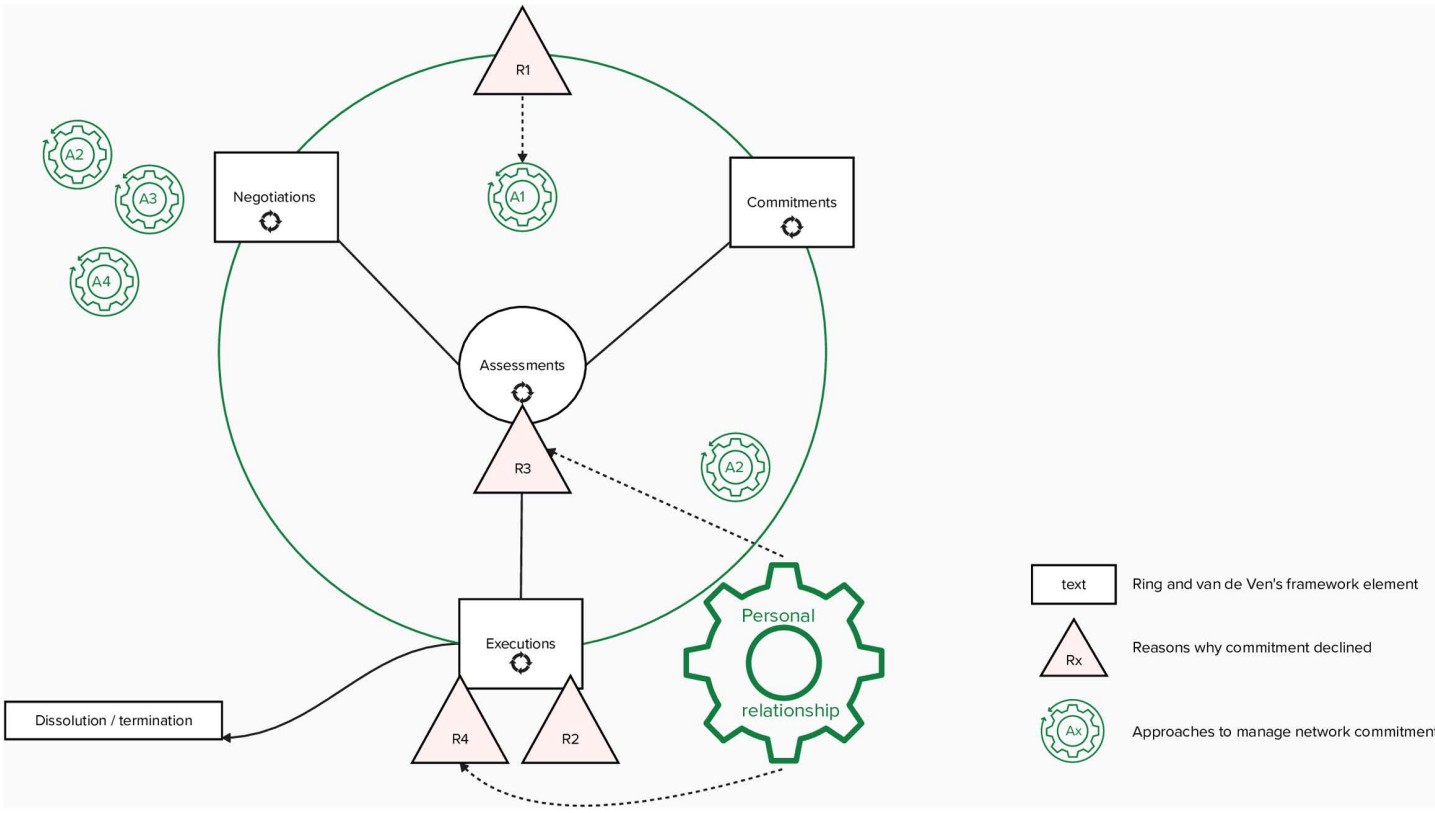

**Fig 1. A dynamic model for network commitment decline and managerial approaches.**

framework, these tensions are positioned between the negotiating and commitments stage and ask for a reflective cycle between negotiation and commitment stages, focusing on assessing goals and needs as one approach to managing commitment(A1).

By emphasizing the importance of strategic timing in managing decline of network commitment we call upon specific managerial actions in one stage in order to be effective in other stages and maintain the commitment. For instance, during the execution stage, a perception of slow progress (R2) may arise. One way to manage this is by highlighting early wins (A2) in the execution stage to counteract perceptions of slow progress, build confidence, and motivate subsequent accomplishments. This could lead to higher commitment [46,47]. The management of expectations (A2) should however begin when joint goals and expectations are established in the negotiation stage. Additionally, lack of reciprocity (R3) especially arises in the execution stage. Bi-directional knowledge sharing is crucial for engagement in healthcare networks. To facilitate reciprocity, network coordinators can establish a set of preconditions (A3), which need to be introduced in the negotiation stage. Lastly, in the execution stage resistance to change (R4) can arise at the organizational level when several collaborative cycles have passed. Involving board-level leadership and securing mandates (A4) in the negotiation stage are helpful. In the negotiation stage, network coordinators should assess whether the most relevant stakeholders are involved, as this is key for effective network functioning. Through these examples, the adjusted Ring and Van de Ven's framework [43] demonstrates that managing network commitment in healthcare improvement networks is not a linear process, but rather one of anticipating challenges across different stages.

### Interdependencies

In addition to mapping the reasons for network commitment to decline and the corresponding managerial approaches across the stages, our study also expands on the role of personal relationship building over time. While the original Ring and Van de Ven's framework [43] emphasizes that repeated cycles of negotiation, commitments, and execution strengthen interpersonal relationships, this study shows how this process is as a turning wheel, activated through ongoing collaboration.

Our figure pictures repeated cycles of negotiation, commitment, and execution in which inter-personal relationships and trust-building play a role in mitigating two sources of declining network commitment: resistance to change at the organizational level and a lack of reciprocity. Participants are more likely to share information and knowledge when they trust each other, and information sharing is not a one-way effort. Conversely, when participants perceive a lack of reciprocal benefits, especially over time, they tend to disengage, share less, and gradually lose trust, leading to reduced commitment to the network [48]. However, such relationships do not develop instantly as they must be built gradually through repeated collaborative cycles and require active engagement from participants throughout [43]. As such, personal relationships are both an outcome of ongoing interaction and a critical condition for preventing network commitment from declining. This feature underscores the interdependent nature of managing network commitment in healthcare networks.

### Limitations

One limitation is that our study did not comprehensively explore the various levels at which network commitment operates. Existing literature indicates that commitment can manifest at the individual, organizational or network level [49]. While some of our respondents shared their experience of commitment at different levels, our analysis did not delve into how these multiple levels of commitment interact or influence one another. Future research should focus on understanding the interplay between commitment at the individual, organizational, and network levels. Such insights will provide a better view of network dynamics and enable even further the development of targeted network strategies.

### Conclusion

Our study focused on how network coordinators of Dutch healthcare improvement networks managed the network commitment of participants. Using an exploratory design we have identified why network commitment declines and which approaches network coordinators used to manage this. We have incorporated these reasons and approaches into Ring and van de Ven's framework [43] emphasizing the importance of strategic timing of managerial approaches and the interdependency with social mechanisms. Additionally, we have visualized how personal relationships are both an outcome of ongoing networks and a critical condition for preventing network commitment from declining. These insights provide both a theoretical contribution as actionable recommendations for network managers.

### Supporting information

**S1 Appendix. COREQ.**
(PDF)

**S1 Table. Network characteristics.**
(XLSX)

**S2 Appendix. Topic Guide.**
(PDF)

**S3 Appendix. Code tree.**
(PDF)

**S4 Appendix. Code book (in Dutch).**
(XLSX)

## Acknowledgments

The authors would like to express their appreciation to the members of the National Patient Safety Program "Time to Connect" for their contributions to this study.

## Author contributions

**Conceptualization:** Manon Roest, Doris van der Smissen, Anne Marie Weggelaar-Jansen.

**Data curation:** Manon Roest, Doris van der Smissen.

**Formal analysis:** Manon Roest, Doris van der Smissen, Anne Marie Weggelaar-Jansen.

**Funding acquisition:** Anne Marie Weggelaar-Jansen.

**Investigation:** Manon Roest, Doris van der Smissen, Anne Marie Weggelaar-Jansen.

**Methodology:** Manon Roest, Doris van der Smissen.

**Project administration:** Manon Roest, Anne Marie Weggelaar-Jansen.

**Resources:** Doris van der Smissen, Anne Marie Weggelaar-Jansen.

**Supervision:** Doris van der Smissen, Remco Mannak, Leon Oerlemans, Anne Marie Weggelaar-Jansen.

**Validation:** Manon Roest, Doris van der Smissen, Remco Mannak, Leon Oerlemans.

**Visualization:** Manon Roest, Anne Marie Weggelaar-Jansen.

**Writing – original draft:** Manon Roest, Doris van der Smissen, Remco Mannak, Leon Oerlemans, Anne Marie Weggelaar-Jansen.

**Writing – review & editing:** Manon Roest, Remco Mannak, Leon Oerlemans, Anne Marie Weggelaar-Jansen.

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
