## [Decision Letter · Decision Letter 0]

Dear Dr. Roest,

Thank you for submitting your manuscript to PLOS ONE. After careful consideration, we feel that it has merit but does not fully meet PLOS ONE’s publication criteria as it currently stands. Therefore, we invite you to submit a revised version of the manuscript that addresses the points raised during the review process.

We look forward to receiving your revised manuscript.

Kind regards,

Ali B. Mahmoud, Ph.D.

Academic Editor

PLOS ONE

**Journal Requirements:**

Reviewers' comments:

Reviewer's Responses to Questions

**Comments to the Author**

1. Is the manuscript technically sound, and do the data support the conclusions?

Reviewer #1: Partly

Reviewer #2: Partly

2. Has the statistical analysis been performed appropriately and rigorously?

Reviewer #1: N/A

Reviewer #2: N/A

3. Have the authors made all data underlying the findings in their manuscript fully available?

Reviewer #1: No

Reviewer #2: No

4. Is the manuscript presented in an intelligible fashion and written in standard English?

Reviewer #1: Yes

Reviewer #2: No

**Reviewer #1:**  Sustaining the spark: essential strategies for long-term commitment in healthcare networks

Thank you for the opportunity to review this paper on commitment for healthcare workers.

The paper has an interesting aim: how do healthcare networks with

61 long-term quality improvement goals manage to keep participants committed?

Here are my comments

1. In method section, please define criteria for inclusion of healthcare networks

2. Please give a more in dept description of the included healthcare networks. Was it nurses or doctors, nurses or others? Or a mix.

3. The method section is not comprehensive please expand

4. Please include more information about the pilot

5. Definition on network and how you identify that about active healthcare need to be expaned. Perhaps a figure to visualize this?

6. What is an extended network please define

7. How was the themes defined?

8. How can healthcare workers use this paper – consider to include that in the discussion. And what kind of healthcare worker?

**Reviewer #2:**  The topic is highly interesting and the qualitatve research is an appropriate design to explore in depth the inner working mechanisms of how a health network by be sustained from the perspective of actors, however, in general, the paper lack in depth description and conceptual clarification to make sense of the differents social mechanisms identified (trust, reciprocity, stakeholders' involvement, alignement of goals and so on). Please see some references that may or may not be helpful in addressing some conceptual clarity on the complex intertwined relationships underlying the long term commitment of actors in healthcare networks!

1. Abstract

Background : Insufficient specification of the rationale of the study, which seems like a single research question done in non specified context which is necessary to make sense of the research question . The research questions need to stem from the problem specification and a research gap.

Material and methods : No mention of the methods, data collection tools, sites, and qualitative data analysis used?

Results : What do interorganizational healthcare networks with long-term improvement goals stem from? This was not specified and clarified in the background. The results section needs to answer the question about the drivers of the maintenance of long-term commitment.

The results seem vague and do not relate to the question raised: What are the drivers of the maintenance of long-term commitment?

Discussion : The discussion asserts that the study addresses why and how networks wane. This does not align with the research question raised! And this was not addressed, at least in the results section of the manuscript. The recommendations, as well as the policy and practical implications, are rather vague. What does a nuanced approach mean?

Recommendation : Please make sure the research question raised aligns with the results presented and the practical and policy implications. Stating that it is interdependent is not a real insight, as you mention that healthcare networks are complex and address complex wicked challenges. The definition of complexity includes non-linearity, interdependency, and emergence, among other characteristics.

Introduction section

No pagination of the manuscript.

Page 9, line 33 : The framing of sentences needs some English editing.

"Given the complexity of the problems?" What problems?

The competitiveness of goals is not a problem; it is the very nature of healthcare organizations, which have always had paradoxical goals (see Parsons' social systems and Cameron and Quinn’s competitive culture framework).

No rationale underlying the study of network commitment. The objective in the abstract states the exploration of the drivers of the maintenance of long-term commitment. The first paragraph states the relationship between network commitment and care quality, which is a different objective.

The introduction of the Dutch network is abrupt. A careful introduction to the Dutch health system is needed before addressing healthcare network challenges and then the objective of the study.

Paragraph, line 45 : The relationship between commitment and trust is bidirectional, not linear. So the authors might use a less prescriptive affirmation: "Clearly, network commitment is essential to network processes and goal attainment."

Paragraph, lines 55-56 : The research gap is not well defined, and the reference used to justify the research gap is a cross-sectional study on the perceived importance of healthcare networks among healthcare alliances in the US. More systematic reviews and scoping reviews are needed to justify this specific research gap, as well as contextual evidence from Europe and the Dutch context in particular.

What motivated the research? What practical challenges does the Dutch health(care) system face?

Examples can be:

van der Weert, G., Burzynska, K., & Knoben, J. (2022). An integrative perspective on interorganizational multilevel healthcare networks: a systematic literature review. BMC Health Services Research, 22(1), 923.

BELRHITI, Z., BIGDELI, M., LAKHAL, A., DIB, K., ZBIRI, S., & BELABBES, S. (2024). Unravelling collaborative governance dynamics within healthcare networks: a scoping review. Health Policy Planning (in press).

CARMONE, A. E., KALARIS, K., LEYDON, N., SIRIVANSANTI, N., SMITH, J. M., STOREY, A., & MALATA, A. (2020). Developing a common understanding of networks of care through a scoping study. Health Systems & Reform, 6, e1810921.

DE POURCQ, K., DE REGGE, M., VAN DEN HEEDE, K., VAN DE VOORDE, C., PAUL, G., & EECKLOO, K. (2019). The role of governance in different types of interhospital collaborations: a systematic review. Health Policy, 123, 472-479.

Line 57 : The sentence is unclear. Please ensure the problem statement is clear enough: "Any network seeks sustainable performance!"

The authors use the term healthcare network with long-term quality improvement goals—are there any other forms of healthcare networks? Who has developed this typology? Is this stated in policy documents? In that case, please ensure the Dutch healthcare system's context is well clarified to allow the reader to make sense of what is presented.

COREQ

I have no access to Appendix 1

appendix need to be numerotated accordingly 1 2 and so on not appendix 1 and then appendix A

there need also a pagination

Material and methods

The rationale behind the framing of the research question still needs to be well formulated and sufficiently enriched with contextual data from the Netherlands' healthcare system and how this fits into the policies, organizational, and quality-of-care challenges.

Network commitment needs to be defined—what does this mean? Is it about long-term or short-term commitment? Then, of whom? Partners? Healthcare providers? Managers? Health professionals?

Only in the Material and Methods section do the authors state that this research is a commissioned project. This needs to be stated in the Introduction section after the contextual specification and a rich description of the healthcare system where this study is undertaken.

"Data collection" rather than "data gathering"!

What does long-term objectives mean for the selection of cases?

What is the unit of analysis?

What is the case (see Yin, 2018, definition of a case)?

What does active healthcare networks mean? How was this assessed? Is there any database of key performance indicators underlying the purposive sampling?

In purposive sampling, what are the selection criteria? What is the principle used in qualitative sampling (deviant cases? Positive cases? Negative cases?) (see Patton, 2001).

I have no access to Appendix A! Why is the number of the appendix "1" and then "A"? The uploaded manuscript does not include appendices! I am not able to check the interview guide!

What is obligatory involvement? Were the healthcare networks mandatory or voluntary? These aspects need to be well defined in the Introduction!

Line 121 : The verbatim rather states the need to align motivation with why—the underlying motives for engagement—which was not expressed in the title of this subsection!

Line 158 : The phenomenon of resistance to change is not sufficiently expressed in operational terms. It is rather described theoretically. Qualitative studies are context-rich, so what are the manifestations of this resistance to change?

Quote P6a is not really related to resistance to change (the subsection under which it is mentioned). Yet, it rather addresses the strategies to increase the long-term commitment of network partners!

Once again, the categories and themes presented do not align with the research question: drivers for the maintenance of long-term commitment of network partners. Yet, in section line 179 (reciprocity and trust), it is rather about the motivation to join the network!

There should be a framework used to align the phases of network formation and maintenance to organize the themes into a clear and fluent logical account and narrative.

Yet, the presentation of themes still needs to be reorganized to tell a story rather than a succession of initial coding results!

Line 197 : Participants’ or stakeholders’ engagement as a strategy for long-term engagement is already stated in the previous paragraph. If this is important, reorganize under a subsection (see Emerson’s collaborative governance framework).

Discussion

The study results did not actually answer the research question: How and why are healthcare networks sustained? Or how is long-term commitment ensured?

Identifying the reasons/contextual conditions reducing long-term commitment is not the same as the reasons underlying the maintenance of healthcare networks!

What is meant by a nuanced approach?

The authors need to shed light on the interorganizational and intraorganizational contexts that have not been sufficiently described to allow adequate sense-making of the assumed hypotheses of why healthcare networks fail to maintain long-term commitment.

The interdependency paragraph needs to use some frameworks to make sense of the collaborative governance complex social dynamics.

Reflectivity of the researchers needs to be included in the reporting, following the COREQ guidelines!

**Do you want your identity to be public for this peer review?** For information about this choice, including consent withdrawal, please see our Privacy Policy

Reviewer #1: No

Reviewer #2: No

---

## [Author Response · Author response to Decision Letter 1]

17 Apr 2025

Dear reviewer,

We want to thank you for taking the time to evaluate our paper and for recognizing the interesting aim of our study. We also like to thank you for your thorough and insightful comments, which have greatly contributed to enhancing the quality of our paper. Over the past weeks, we have carefully refined, restructured and clarified various sections to strengthen our arguments. Several comments from you and the other reviewers touched on the issue of alignment and overlap. Therefore, we chose to synthesize the feedback, resulting in substantial changes throughout the manuscript. We focused on enhancing the clarity, coherence and alignment of our arguments. One key improvement from our point of view was the integration of a theoretical framework in the discussion section. This helped to structure and visualize our findings. We believe that it improved the overall readability and quality of the manuscript and hope the revisions align with your expectations.

We have provided a response letter as file with a detailed response to each of the reviewers’ comments.

---

## [Decision Letter · Decision Letter 1]

A dynamic model to sustain the spark: How do network coordinators in Dutch healthcare networks maintain network commitment?

PONE-D-24-32664R1

Dear Dr. Roest,

We’re pleased to inform you that your manuscript has been judged scientifically suitable for publication and will be formally accepted for publication once it meets all outstanding technical requirements.

Kind regards,

Ali B. Mahmoud, Ph.D.

Academic Editor

PLOS ONE

Additional Editor Comments (optional):

Reviewers' comments:

Reviewer's Responses to Questions

**Comments to the Author**

Reviewer #1: All comments have been addressed

2. Is the manuscript technically sound, and do the data support the conclusions?

Reviewer #1: Yes

3. Has the statistical analysis been performed appropriately and rigorously?

Reviewer #1: N/A

4. Have the authors made all data underlying the findings in their manuscript fully available?

Reviewer #1: Yes

5. Is the manuscript presented in an intelligible fashion and written in standard English?

Reviewer #1: Yes

Reviewer #1: The authors have made through adjustments to the revision.

Comments have been addressed well.

No further comments.

**Do you want your identity to be public for this peer review?** For information about this choice, including consent withdrawal, please see our Privacy Policy

Reviewer #1: No

---

## [Editor Report · Acceptance letter]

PONE-D-24-32664R1

PLOS ONE

Dear Dr. Roest,

I'm pleased to inform you that your manuscript has been deemed suitable for publication in PLOS ONE. Congratulations! Your manuscript is now being handed over to our production team.

Kind regards,

on behalf of

Dr. Ali B. Mahmoud

Academic Editor

PLOS ONE